# Mediterranean Diet Adherence, Body Mass Index and Emotional Intelligence in Primary Education Students—An Explanatory Model as a Function of Weekly Physical Activity

**DOI:** 10.3390/children9060872

**Published:** 2022-06-11

**Authors:** Eduardo Melguizo-Ibáñez, Gabriel González-Valero, Georgian Badicu, Ana Filipa-Silva, Filipe Manuel Clemente, Hugo Sarmento, Félix Zurita-Ortega, José Luis Ubago-Jiménez

**Affiliations:** 1Faculty of Education Sciences, Department of Didactics of Musical, Plastic and Corporal Expression, University of Granada, 18071 Granada, Spain; edumeliba@correo.ugr.es (E.M.-I.); felixzo@ugr.es (F.Z.-O.); jlubago@ugr.es (J.L.U.-J.); 2Faculty of Education and Sport Sciences (Melilla), Department of Didactics of Musical, Plastic and Corporal Expression, University of Granada, 52071 Melilla, Spain; ggvalero@ugr.es; 3Department of Physical Education and Special Motricity, Faculty of Physical Education and Mountain Sports, Transilvania University of Braşov, 500068 Braşov, Romania; 4Escola Superior Desporto e Lazer, Instituto Politécnico de Viana do Castelo, Rua Escola Industrial e Comercial de Nun’Álvares, 4900-347 Viana do Castelo, Portugal; anafilsilva@gmail.com (A.F.-S.); filipe.clemente5@gmail.com (F.M.C.); 5Research Center in Sports Performance, Recreation, Innovation and Technology (SPRINT), 4960-320 Melgaço, Portugal; 6The Research Centre in Sports Sciences, Health Sciences and Human Development (CIDESD), 5001-801 Vila Real, Portugal; 7Instituto de Telecomunicacoes, Delegacao da Covilha, 1049-001 Lisboa, Portugal; 8University of Coimbra, Research Unit for Sport and Physical Activity (CIDAF), Faculty of Sport Sciences and Physical Education, 3040-248 Coimbra, Portugal; hg.sarmento@gmail.com

**Keywords:** Mediterranean diet, emotional intelligence, physical activity, body mass index

## Abstract

Adolescence is a key developmental period from a health, physical and psychological perspective. In view of this, the present research aimed to establish the relationship between emotional intelligence, Mediterranean diet adherence, BMI and age. In order to address this aim, (a) an explanatory model is developed of emotional intelligence and its relationship with Mediterranean diet adherence, BMI and age, and (b) the proposed structural model is examined via multi-group analysis as a function of whether students engage in more than three hours of physical activity a week. To this end, a quantitative, non-experimental (ex post facto), comparative and cross-sectional study was carried out with a sample of 567 students (11.10 ± 1.24). The instruments used were an ad hoc questionnaire, the KIDMED questionnaire and the TMMS-24. Outcomes reveal that participants who engage in more than three hours of physical activity a week score more highly for emotional intelligence than those who do not meet this criterion. Furthermore, it was also observed that, whilst the majority of the sample was physically active, improvement was required with regards to Mediterranean diet adherence.

## 1. Introduction

In a physical and psychological sense, adolescence marks a sensitive growth phase [1], which starts during puberty and finishes with bio-psychosocial maturity [2]. It typically takes place between 10 and 19 years of age [3]. With regards to psychological aspects, adolescents are yet to fully configure their personality, leading them to be easily influenced [3]. Thus, it is important to encourage them to adopt positive habits with regards to physical activity and health [4]. Whilst the physical and psychological benefits of an active and healthy lifestyle are clear, during adolescence, a reduction is seen in the time spent engaged in physical activity and the quality of dietary patterns [5]. 

The Mediterranean diet is considered to be a healthy dietary pattern, not only in terms of the quantity of food intake but, also, in terms of quality, food preparation and nutrient suitability [6]. Currently, a change in dietary patterns is taking place in Western societies, with an increase in pre-cooked dishes with a high caloric level, increasing the incidence of cardiovascular diseases. [7]. Food availability is one of factors leading to a worsening of dietary habits [8]. The Mediterranean diet is characterized by the intake of specific food varieties, especially wholegrains, olive oil, dairy products, bread, fruits and vegetables [9]. Caprara [10] also revealed that Mediterranean diet adherence combined with high levels of physical activity helped to increase life expectancy [11]. Positive adherence to a dietary pattern also brings numerous benefits in physical and mental areas [7]. In the physical area, a reduction in BMI, a reduction in waist circumference and the prevention of cardiovascular diseases are observed [11], whereas in the mental area, notable improvements in self-concept and emotional intelligence are observed [7]. 

Physical activity has been defined as any exercise performed by the skeletal muscles involving energetic expenditure [12]. Villasana et al. [13] showed that weekly physical activity engagement falls during adolescence. Likewise, Hernández-Álvarez et al. [14] state that this drop in physical exercise is due to a greater preference for sedentary habits over active activities. WHO [15] states that, in order to follow an active lifestyle, children and adolescents aged 5 to 17 years old should engage in moderate and vigorous physical activity for at least 60 min a day on three days a week. Following these guidelines brings physical health benefits such as a reduction of waist circumference, improvement of blood pressure and reduced BMI, as well as numerous improvements to mental health, such as better emotional control [16,17]. 

Emotional control plays a key role in leading a healthy lifestyle [18]. It can be defined as a multidimensional experience with three response systems: cognitive/subjective, behavioural/expressive and physiological/adaptive [19]. Further, educational settings must provide emotional education as a key element to the integral development of adolescents [20]. Thus, within the classroom, emotional intelligence is conceived as a key element to emotional control [21]. Among the different definitions proposed for the concept of emotional intelligence, the most widely accepted is that proposed by Salovey et al. [19]. These authors state that emotional intelligence describes the ability to accurately perceive, value and express emotions in a personal way, whilst also understanding the emotions experienced by others. Moreover, this model outlines three stages that make up emotional intelligence. The first of these is emotional perception, which pertains to the ability to perceive one’s own emotions and others’ emotions [22]. The second, emotional understanding, corresponds to the ability to understand and process information garnered from emotions [19] and emotional regulation and the ability to promote understanding and personal growth [19]. In this case during adolescence, emotional intelligence plays a fundamental role in overcoming disruptive states [20]; however, when adulthood is reached, the construct that makes up emotional intelligence (attention, clarity and repair) helps people to become emotionally competent [18,22]. Finally, another factor that can affect the development of emotional intelligence is the mental image perceived by the subjects of themselves [7]. In this case, it has been observed that people who are overweight show worse levels of emotional intelligence than those who reflect a normal weight and are satisfied with their physical appearance [17].

Based on the above, the present research outlines the following hypotheses: -Participants who report more than three hours of physical activity a week will reveal better associations between BMI, Mediterranean diet adherence and emotional intelligence. -Participants who report engaging in less than three hours of physical activity a week will reflect worse associations between BMI, Mediterranean diet adherence and emotional intelligence. -Participants who engage in more than three hours of physical activity a week will have better scores for emotional intelligence, Mediterranean diet adherence and BMI than those who do not claim to meet these criteria. 

In conclusion, the present research aims to establish the relationship between emotional intelligence, Mediterranean diet adherence, BMI and age. In order to address this aim, (a) an explanatory model of emotional intelligence and its relationship with Mediterranean diet adherence, BMI and age will be developed, and (b) the structural model will be tested via multi-group analysis as a function of whether students engage in more than three hours of physical activity a week.

## 2. Materials and Methods

### 2.1. Participants and Design

A non-experimental (ex post facto), descriptive and cross-sectional study was carried out with students from different public (state-funded) schools in the city of Granada. Schools were selected to participate randomly. The sample was composed of a total of 567 students aged between 9 and 13 years old (11.10 ± 1.24). With regards to sex, 53% were male (*n* = 303) and 47% were female (*n* = 264). Data collection was carried out after obtaining consent from legal guardians. A cover letter was sent to parents informing them about the aims and nature of the study and, subsequently, written informed consent was obtained. In terms of sample size, the overall sample of primary school students in the city of Granada is 83,940. Setting a 5.0% sampling error gave a confidence level of 97% for the recruited sample size. 

### 2.2. Instruments

Sociodemographic questionnaire: Designed to collect information on age and sex. This tool was also used to collect data on sport or physical activity engagement. Participants were asked whether or not they engaged in more than 3 h of physical activity a week (do you engage in more than 3 h of physical activity outside of school hours?). In this case, this question is used to check whether the participants are physically active or not [23]. Participating Physical Education teachers measured and weighed participants for subsequent calculation of BMI. In this case, the mathematical formula of dividing the participant’s weight by the squared height was used to calculate the BMI. The cut-off points used in the present investigation are the specific values for children according to their age that correspond to the following values for adults: Obesity (scores above 30 kg/m^2^), overweight (scores between 25 kg/m^2^ and 29.99 kg/m^2^), normal weight (values between 24.99 kg/m^2^ and 18 kg/m^2^), underweight (scores below 18.5 kg/m^2^) and severely underweight (less than 16.5 kg/m^2^) [24,25].

KIDMED questionnaire: The Spanish version developed by Serrá-Majem et al. [26] was used for the present study. This questionnaire includes a total of 16 items that can be answered negatively or positively. Items 5, 11, 13 and 15 are negatively framed and, therefore, positive responses are scored as −1. All remaining questions are negatively framed, with affirmative responses being scored as +1. Potential final scores range from 4 to 12. Scores are then classified according to three groups describing Mediterranean diet adherence. The following groups are defined: optimal diet (8 and above), needs improvement (2–7) and poor diet (1). Reliability analysis of the data obtained for this questionnaire yielded acceptable results (α = 0.771).

Trait Meta-Mood Scale 24: Developed by Salovey et al. [19]. The Spanish version developed by Fernández-Berrocal et al. [27] was used in the present study. In this case, the version used for data collection is adapted for the study population [27]. This questionnaire comprises a total of 24 items that are rated on a five-point Likert scale (1 = “Disagree” to 5 = “Strongly agree”). On this scale, emotional intelligence is evaluated as a construct made up of three factors. These factors are emotional attention (items 1, 2, 3, 4, 5, 6, 7 and 8), understanding of emotional states (items 9, 10, 11, 12, 13, 14, 15 and 16) and emotional regulation (17, 18, 19, 20, 21, 22, 23 and 24). Items are summed to produce an overall score. In the present study, reliability indices for emotional attention (α = 0.902), emotional clarity (α = 0.845) and emotional repair (α = 0.891) were all acceptable.

### 2.3. Procedure

During data collection, an information pack was sent out by the Department of the Didactics of Musical, Artistic and Corporal Expression of the University of Granada. This letter informed school management about the aims and scope of the study. Following this, an e-mail was sent by the schools to students’ legal guardians to inform them of the study aims and request their consent for participation of their children in the research project. Once this permission was obtained, schools were provided with a link to a Google Form which laid out the previously described aims. Due to the COVID-19 pandemic, the project investigators were unable to personally attend the school, as during data collection (March 2021), entry to schools was restricted. Teachers received instructions from researchers around how questionnaires should be completed. Further, two questions were intentionally repeated on the questionnaire in order to check for random responding and ensure that questionnaires were completed correctly. As a result, a total of 19 questionnaires were discarded due to incorrect completion. Finally, the present study complied with the ethical standards for research involving human subjects set out in the Helsinki Declaration of 1975 and was supervised by the Research Ethics Committee of the University of Granada (1230/CEIH/2020).

### 2.4. Data Analysis

The IBM SPSS Statics 25.0 program (IBM Corp, Armonk, NY, USA) was used for data analysis. A descriptive analysis was carried out according to frequencies. Comparative analysis was performed using contingency tables and Student t-tests for independent samples. Statistically significant differences were determined using Pearson’s Chi-square statistics, with a significance level of 95%. In order to calculate the statistical power achieved for t-statistic calculation, Cohen’s standardized d-index was used [28]. In this sense, effects are interpreted as null (≤0.19), small (0.20–0.49), medium (0.50–0.79) and large (≥0.80). Phi and Cramer’s V was used for interpretation of contingency table outcomes, with effects being interpreted as weak (<0.2), moderate (0.2–0.59) and large (≥0.6) [29]. The Kolmogorov-Smirnov test was used to examine data distribution, with a normal distribution being confirmed.

The IBM SPSS Amos 26.0 program (IBM Corp, Armonk, NY, USA) was used to develop structural equation models. This model allows existing relationships to be established between participants who engage in more than 3 h of weekly physical activity and those who do not meet this physical-sport criterion. In this case, each model is composed of three endogenous variables (MDA; AGE; BMI) and three exogenous variables (EA; EC; ER). For the endogenous variables, causal relationships are examined in consideration of observed associations between the indicators and the degree of measurement reliability. This enables the error originated by the measurement of observed variables to be included in the model. Figure 1 was developed in which unidirectional arrows are produced from regression weights and represent lines of influence. In addition, a significance level of 0.05 was established. In this case, relationships between the three-factor construct of emotional intelligence and age, body mass index and Mediterranean diet adherence were examined.

Finally, model fit was evaluated following visualisation of its different parameters. According to Bentler [30] and McDonald [31], goodness of fit should be evaluated according to chi-square, with non-significant *p*-values indicating good fit. With regards to the comparative fit index (CFI), values above 0.95 reflect good fit. With regards to the goodness of fit index (GFI), scores above 0.90 show good fit. With regards to the incremental reliability index (IFI), values above 0.90 reflect good fit. Finally, in the case of root mean square approximation (RMSEA), scores below 0.1 evidence adequate model fit. 

## 3. Results

With regards to emotional attention, the outcomes presented in Table 1 show that higher scores were recorded for individuals who met the established the sport criteria (M = 3.496) relative to participants who did not meet this criterion (M = 3.423). Turning attention to emotional clarity, higher scores were reported by participants who engaged in more than 3 h of physical activity a week (M = 3.612) relative to those who spent less time engaged in such activity (M = 3.429). Finally, in terms of emotional repair, higher scores were observed for participants who met the sport criterion (M = 3.792) relative to those who did not (M = 3.423).

Table 2 shows the descriptive analysis for the BMI variable. In this case, it is observed that 29.6% (*n* = 168) show as underweight, whereas more than two thirds of the total show as normal weight (69.1%; *n* = 392). Only 1.3% were overweight (*n* = 7).

Results presented in Table 3 (*p* ≤ 0.05) reveal that participants who did engage in physical activity were more likely to be underweight (*n* = 156; 31.3%) than overweight (*n* = 12; 2.4%). In the case of normal weight, higher scores are observed for those who did not engage in more than 3 h of physical activity a week (*n* = 57; 82.6%).

Table 4 pertains to the relationship between physical activity and Mediterranean diet adherence, with statistically significant differences emerging at the level *p* ≤ 0.05. In this case, it is observed that participants who met the physical activity/sport criterion (*n* = 159; 89.8%) were more likely to follow an optimal diet than those who did not (*n* = 18; 10.2%).

The structural equation models (Figure 2) demonstrated good fit for each of the different examined indices. Chi-square analysis produced a significant *p*-value (X^2^ = 1.611; df = 3; pl = 0.657); however, these data cannot be interpreted in isolation due to the sensitivity of this statistic to sample size [32]. For the general model, other standardised fit indices were used which are less sensitive to sample size. Comparative fit (CFI), normalised fit (NFI) and incremental fit (IFI) indices were 0.998, 0.994 and 0.998, respectively, whereas the Tucker-Lewis index (TLI) was 0.946. All of these indices demonstrate excellent fit. In addition, the root mean of square error of approximation (RMSEA) value obtained was 0.010.

In this case, Figure 2 and Table 5 show the relationships between the variables for the whole sample. A positive relationship is observed between age and emotional attention (r = 0.441), whereas negative relationships are observed for emotional clarity (r = −0.059) and emotional repair (r = −0.004). Continuing with the relationship between the emotional domain and adherence to the Mediterranean diet, a positive relationship was observed with all the variables that make up emotional intelligence (r = 0.066; r = 0.046), showing a level of significance with emotional repair (*p* ≤ 0.05; r = 0.111). Finally, body mass index shows a positive relationship with emotional clarity (*p* ≤ 0.05; r = 0.143) and emotional attention (r = 0.004) and a negative relationship with emotional repair (*p* ≤ 0.05; r = −0.140). 

Turning attention to the structural equation models, the model developed for participants who meet the physical-sporting criteria produced a significant *p*-value (X^2^ = 6.072; df = 3; pl = 0.108). As with before, the CFI produced was 0.992, whereas NFI and IFI values were 0.986 and 0.993, respectively. A TLI of 0.946 was also obtained, with all of these aforementioned values pointing to excellent fit. Finally, the RMSEA value obtained was 0.045.

Both Figure 3 and Table 6 present the regression weights associated with the theoretical model, with statistically significant relationships being indicated at *p* < 0.05 and *p* < 0.001. In this case, emotional attention was positively related with Mediterranean diet adherence (*p* < 0.001; r = 0.186), age (r = 0.067), emotional clarity (*p* < 0.001; r = 0.482) and emotional repair (*p* < 0.001; r = 0.435), with a negative relationship being found with body mass index (r = −0.067). Moving on to consider emotional clarity, a positive relationship is observed with Mediterranean diet adherence (r = 0.045), body mass index (*p* ≤ 0.05; r = 0.165) and emotional repair; however, a negative relationship is shown with age (r = −0.112). In the case of emotional repair, positive relationships are observed with Mediterranean diet adherence (r = 0.071) and age (r = 0.021), whereas a negative relationship is seen with body mass index (*p* < 0.001; r = −0.209). 

Turning attention to the structural equation models, the model developed to participants who do not engage in more than 3 h showed good fit in terms of all of the different examined indices. Chi-square analysis produced a significant *p*-value (X^2^ = 8.330; df = 3; pl = 0.040). As with before, the CFI produced was 0.955, whereas NFI and IFI values were 0.912 and 0.928, respectively. A TLI of 0.934 was also obtained, with all of these aforementioned values pointing to excellent fit. Finally, the RMSEA value obtained was 0.056.

Figure 4 and Table 7 present the regression weights associated with the theoretical model, with statistically significant relationships being indicated at the level of *p* < 0.05 and *p* < 0.001. In this case, emotional attention was positively related with Mediterranean diet adherence (r = 0.130), body mass index (*p* < 0.05; r = 0.383), emotional clarity (*p* < 0.001; r = 0.464) and emotional repair (*p* < 0.001; r = 0.510), whereas it was negatively related with age (*p* < 0.05; r = −0.289). Emotional clarity produced positive associations with age (*p* < 0.05; r = 0.302) and emotional repair (*p* < 0.001; r = 0.519); however, negative relationships were produced Mediterranean diet adherence (r = −0.045) and body mass index (r = −0.056). Finally, with regards to emotional repair, positive relationships emerged Mediterranean diet adherence (r = 0.129), age (r = 0.041) and body mass index (r = 0.07). 

## 4. Discussion

The present study describes existing relationships between emotional intelligence, adherence to the Mediterranean diet, age and body mass index as a function of the time spent engaged in weekly physical activity. Findings met the proposed objectives and, therefore, the following discussion serves to compare obtained outcomes with those reported in previous research. 

Present findings show that individuals who report engaging in more than three hours of weekly physical activity also report higher scores for attention, clarity and emotional repair than those who engage in such activity for less than three hours. In view of these results, Vaquero-Solís et al. [33] argue that physical activity engagement is beneficial for emotional control. Likewise, Angarita-Ortiz et al. [34] uphold that physical activity engagement contributes to better emotional intelligence, since physical exercise is used as a means to achieving emotional well-being.

With regards to the relationship between BMI and physical activity engagement, it was observed that students who claimed to engage in more than three hours of physical activity a week were more likely to be overweight or underweight than those who did not meet the aforementioned criteria. Hugely contrasting findings were reported by Chavesa et al. [35] and Power et al. [36]. Explanations for this can be found in the study conducted by Salcin et al. [37] who identified during adolescence a lack of control over diet exists, with an increase in fat intake and an increase in the consumption of pre-cooked dishes with a high calorie content. Further, Melguizo-Ibáñez et al. [38] and Padial-Ruz et al. [39] argue that family functioning plays a key role in diet control and encouraging physical activity.

Turning attention to Mediterranean diet adherence and physical activity, it was observed that more than half of participants who engaged in more than three hours of physical activity a week were in need of improving their diet. Similar findings were reported by Melguizo-Ibáñez et al. [40] and Morales-Camacho et al. [41] who found that dietary patterns worsen during adolescence. Moreover, Muros et al. [42] argue that adherence tends to reduce due to the fact that adolescents tend to consume more saturated fats in place of foods rich in proteins and carbohydrates.

With regards to the present associative analysis, students who engaged in more than three hours of physical activity a week were observed to demonstrate a stronger relationship between Mediterranean diet adherence and attention and emotional clarity. Similar outcomes were obtained by Marfil-Carmona et al. [43] who stated that healthy lifestyles in general have a positive impact on the overall development of adolescents, including that of the emotional sphere [44]. On the other hand, with regards to the relationship between Mediterranean diet adherence and emotional repair, participants who engaged in less than three hours of physical activity a week were observed to obtain better outcomes. Contrasting results were obtained by Castro-Sánchez et al. [45] who concluded that engagement in physical exercise helps to channel disruptive states and, consequently, improve emotional outcomes. 

With regards to BMI and emotional intelligence, participants who engaged in more than three hours of physical activity a week demonstrated a negative relationship between BMI, attention and emotional clarity. In view of these findings, Cuesta-Zamora et al. [46] conclude that obsession with physical fitness can lead to a continuous state of body dissatisfaction, leading to a state of non-acceptance at the detriment to the emotional state of adolescents. A positive relationship between BMI and emotional clarity was also observed in participants who engaged in more than three hours of physical activity a week. These findings are supported by Andrei et al. [47] who argue that emotional clarity helps to improve mental health and, as a result, improve physical performance and physical condition.

Outcomes pertaining to emotional intelligence reveal that participants who engage in more than three hours of physical activity a week demonstrate better outcomes. In this sense, Vaquero-Solís et al. [33] conclude that emotional clarity, together with the attention given to the negative feelings experienced during physical activity, helps to channel disruptive emotions due to the division of neurotransmitters [48]. In contrast, better relationships between emotional attention and repair were shown in participants who do not engage in more than three hours of physical activity a week. Contrasting findings have been reported by Wang et al. [49] who stated that physical activity engagement provides benefits for the perception of feelings and coping. Further, research by Trigueros et al. [50] concluded that an active lifestyle helps to improve emotional and physical well-being.

Finally, the present research reports that leading an active lifestyle brings benefits in the emotional and physical area [51]. In this case, the present study highlights the need to create a motivation and positive attitude towards the practice of physical activity at an early age [50] that will last into adulthood and help to improve people’s health [52] and the decrease of cardiovascular diseases in different societies [53]. 

### Limitations and Future Perspectives

The present study has a number of limitations. The main limitation is that the present study is cross-sectional and based on a single measurement in a specific population at a determined timepoint. This only allows for the identification of relationships between variables at the examined timepoint and prevents causal relationships from being established. Another limitation is that the sample consisted of primary school pupils from a very specific geographical area. This prevents findings from being generalised to wider geographical areas at a national or regional level. In addition, the presence of COVID-19 and the resultant mobility restrictions imposed had an impact on data collection, as researchers were unable to access schools, leading to a smaller sample being recruited. It should also be noted that, although reliable instruments were used to collect weight and height data, these instruments carry an implicit measurement error. Despite recruiting a sizeable final sample which enabled a confidence level of 97%, it was not possible to collect data from other schools. As a result, external variables such as socioeconomic level, education or school ownership could not be considered.

In terms of future perspectives, it would be interesting to carry out a comprehensive experimental study targeting nutritional and emotional education, via physical activity engagement.

## 5. Conclusions

Descriptive analysis showed that participants who engaged in more than three hours of physical activity a week have better emotional intelligence than those who do not meet this criterion. Furthermore, it was also observed that the majority of the sample was physically active; however, improvements were required with regards to Mediterranean diet adherence. 

Turning attention to the structural equation models, better relationships between Mediterranean diet adherence and attention and emotional clarity were observed in participants who reported more than 3 h of physical activity a week; however, better relationships were seen between Mediterranean diet adherence and emotional repair in those who did not achieve this level of physical activity. A better relationship between BMI and emotional repair and attention was also observed in participants who engaged in less than 3 h of physical activity a week. 

Finally, a stronger relationship was observed between the variables comprising emotional intelligence in participants who engaged in more than 3 h of weekly physical activity, whereas participants who do not meet this criterion demonstrated a stronger relationship between emotional attention and emotional repair.

## Figures and Tables

**Figure 1 children-09-00872-f001:**
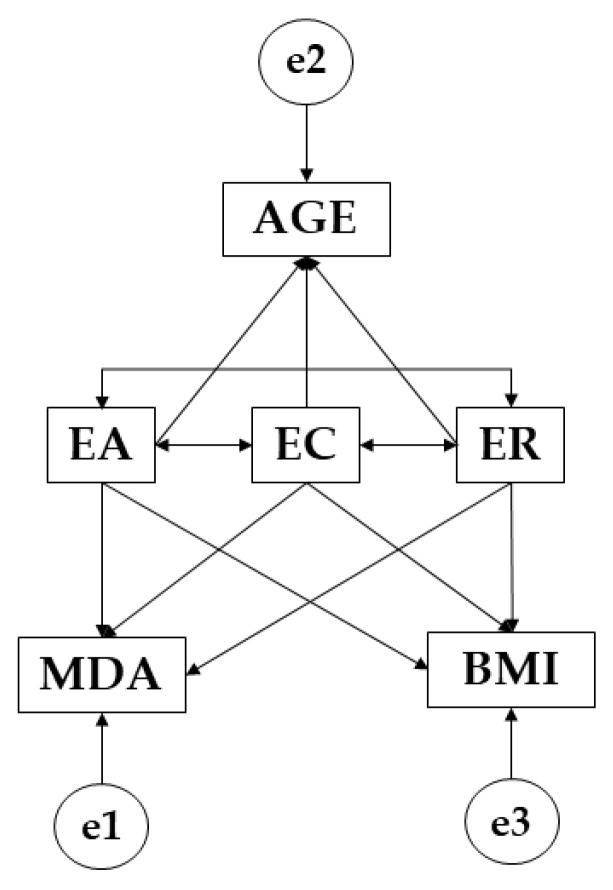
Structural equation model. Note: Emotional attention (EA); emotional clarity (EC); emotional repair (ER); body mass index (BMI); Mediterranean diet adherence (MDA); age (AGE).

**Figure 2 children-09-00872-f002:**
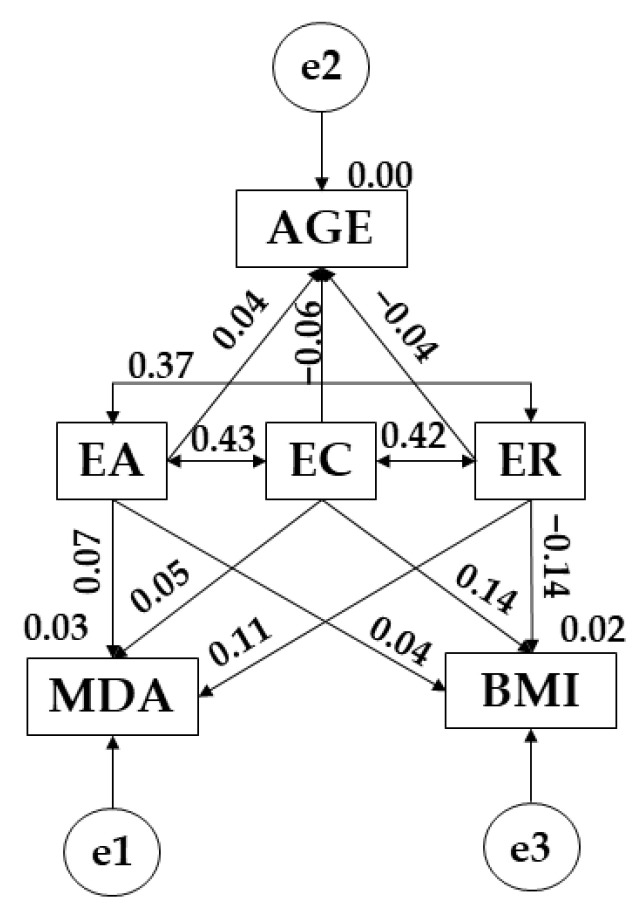
SEM pertaining to the entire sample. Note: Emotional attention (EA); emotional clarity (EC); emotional repair (ER); body mass index (BMI); Mediterranean diet adherence (MDA); age (AGE).

**Figure 3 children-09-00872-f003:**
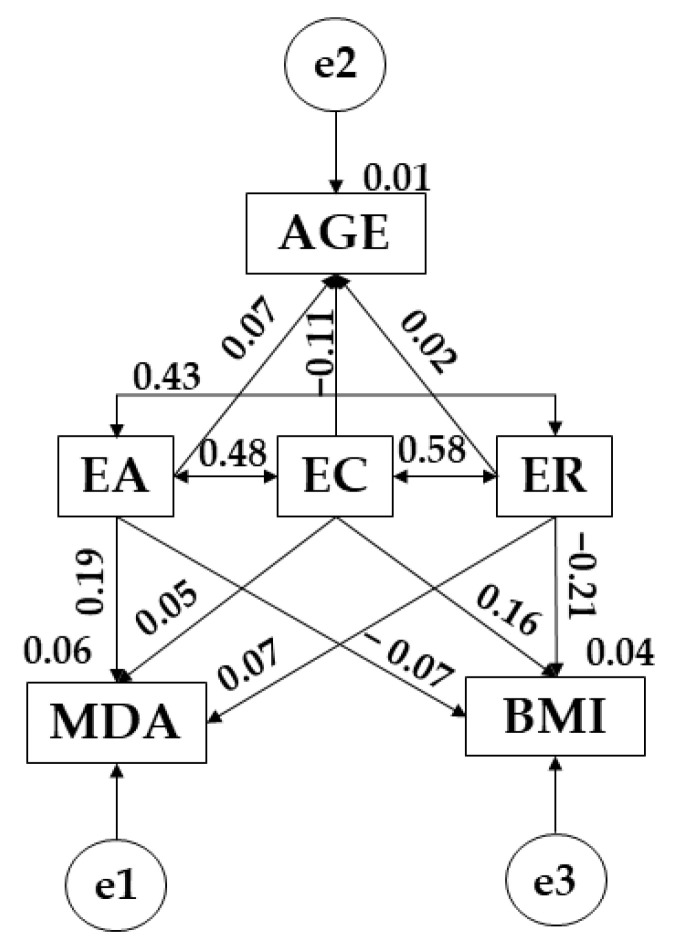
SEM pertaining to participants who engage in more than 3 h of PA a week. Note: Emotional attention (EA); emotional clarity (EC); emotional repair (ER); body mass index (BMI); Mediterranean diet adherence (MDA); age (AGE).

**Figure 4 children-09-00872-f004:**
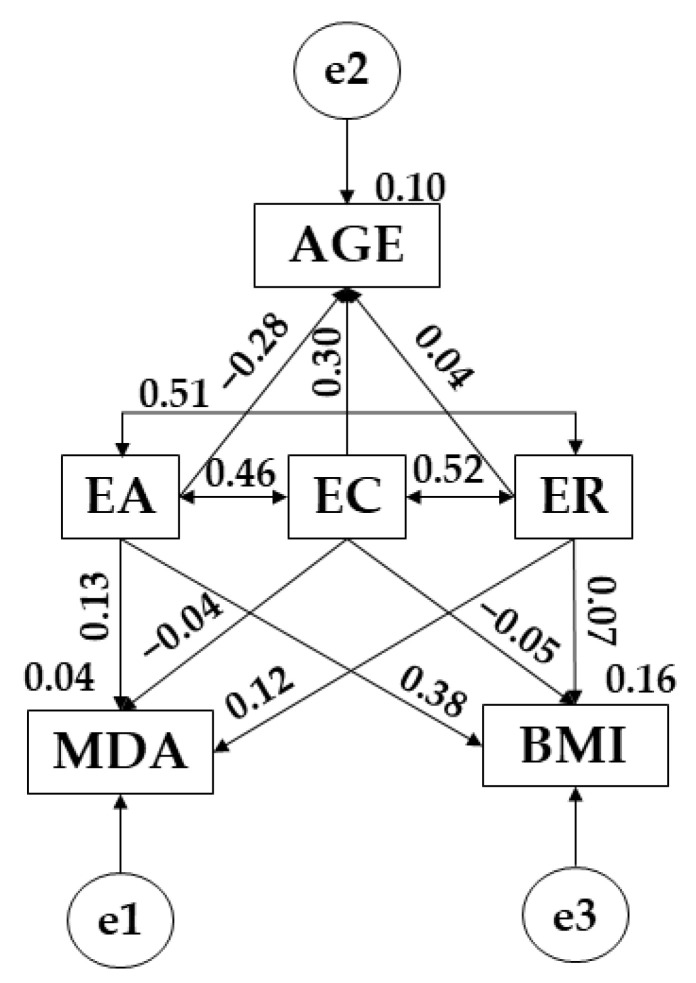
SEM pertaining to participants who do not engage in more than 3 h of PA a week. *Note:* Emotional attention (EA); emotional clarity (EC); emotional repair (ER); body mass index (BMI); Mediterranean diet adherence (MDA); age (AGE).

**Table 1 children-09-00872-t001:** Comparative analysis between physical activity (Yes/No) and emotional intelligence.

		Levene Test					*t*-Test			ES (d)	95% CI
		N	M	SD	F	Sig	T	df	P		
**EA**	**No**	69	3.423	0.825	0.352	0.553	−0.684	86.915	>0.05	0.525	[0.161; 0.342]
	**Yes**	498	3.496	0.805							
**EC**	**No**	69	3.429	0.877	0.009	0.926	−1.628	0.107	>0.05	0.214	[0.038; 0.466]
	**Yes**	498	3.612	0.851							
**ER**	**No**	69	3.423	0.815	0.763	0.383	−3.520	87.380	≤0.05	0.458	[0.205; 0.711]
	**Yes**	498	3.792	0.805							

Note: 1 Emotional attention (EA); emotional clarity (EC); emotional repair (ER). Note 2: Sample (N); Average (M); Standard Deviation (SD); F-Snedecor for equality of variances (F); Significance Level (Sig); T-value (T); Degrees of Freedom (df); Bilateral Significance (P); Cohen’s standardized d-index (ES (d)); 95% Confidence Interval of the Difference (95% CI).

**Table 2 children-09-00872-t002:** BMI descriptive analysis.

		Frequency	Percentage
**Body Mass Index**	**Underweight**	168	29.6%
	**Normal weight**	392	69.1%
	**Overweight**	7	1.3%
**Total**		567	100%

**Table 3 children-09-00872-t003:** Comparative analysis of body mass index and physical activity engagement.

ES	Phi = 0.119		Body Mass Index			Total
	Cramer’s V = 0.119		Underweight	Normal Weight	Overweight	
**PA**	**No**	Count	12	57	0	69
		% PA	17.4%	82.6%	0.0%	100%
	**Yes**	Count	156	330	12	498
		% PA	31.3%	66.3%	2.4%	100%

Note 1: Physical Activity (PA).

**Table 4 children-09-00872-t004:** Comparative analysis of Mediterranean diet adherence and physical activity engagement.

ES	Phi = 0.083		PA		Total
	Cramer’s V = 0.083		No	Yes	
**MDA**	**Poor quality**	Count	3	18	21
		% MDA	14.3%	85.7%	100%
	**Needs improvement**	Count	48	321	369
		% MDA	13.0%	87.0%	100%
	**Optimal diet**	Count	18	159	177
		% MDA	10.2%	89.8%	100%

Note 1: Physical Activity (PA); Mediterranean Diet Adherence (MDA).

**Table 5 children-09-00872-t005:** SEM belonging to the entire sample.

Association between Variables	RW				SRW
	Estimation	SE	CR	*p*	Estimation
AGE ← EA	0.075	0.097	0.771	0.441	0.037
AGE ← EC	−0.115	0.094	−1.218	0.223	−0.059
AGE ← ER	−0.008	0.094	−0.087	0.931	−0.004
MDA ← EA	0.059	0.042	1.406	0.160	0.066
MDA ← EC	0.038	0.041	0.947	0.343	0.046
BMI ← EC	0.109	0.037	2.954	**	0.143
BMI ← ER	−0.110	0.037	−2.975	**	−0.140
MDA ← ER	0.096	0.041	2.366	**	0.111
BMI ← EA	0.003	0.038	0.084	0.933	0.004
EC ← → EA	0.169	0.018	9.332	***	0.426
EC ← → ER	0.170	0.019	9.198	***	0.419
EA ← → ER	0.143	0.017	8.296	***	0.372

Note 1: Regression weights (RW); standardized regression weights (SRW); standard error (SE); critical ratio (CR). Note 2: Emotional attention (EA); emotional clarity (EC); emotional Repair (ER); body mass index (BMI); Mediterranean diet adherence (MDA); age (AGE). Note 3: ** *p* ≤ 0.05; *** *p* < 0.001.

**Table 6 children-09-00872-t006:** SEM pertaining to participants who engage in more than 3 h of PA per week.

Association between Variables	RW				SRW
	Estimation	SE	CR	*p*	Estimation
MDA ← EA	0.032	0.009	3.666	***	0.186
AGE ← EA	0.106	0.082	1.293	0.196	0.067
BMI ← EA	−0.042	0.032	−1.307	0.191	−0.067
MDA ← EC	0.007	0.009	0.808	0.419	0.045
AGE ← EC	−0.166	0.086	−1.937	0.053	−0.112
BMI ← EC	0.097	0.034	2.891	**	0.165
MDA ← ER	0.012	0.010	1.297	0.195	0.071
AGE ← ER	0.033	0.088	0.380	0.704	0.021
BMI ← ER	−0.131	0.035	−3.775	***	−0.209
EC ← → EA	0.330	0.034	9.680	***	0.482
EC ← → ER	0.398	0.036	11.199	***	0.581
EA ← → ER	0.282	0.032	8.888	***	0.435

Note 1: Regression weights (RW); standardized regression weights (SRW); standard error (SE); critical ratio (CR). Note 2: Emotional attention (EA); emotional clarity (EC); emotional Repair (ER); body mass index (BMI); Mediterranean diet adherence (MDA); age (AGE). Note 3: ** *p* ≤ 0.05; *** *p* < 0.001.

**Table 7 children-09-00872-t007:** SEM pertaining to participants who do not engage in more than 3 h of PA a week.

Association between Variables	RW				SRW
	Estimation	SE	CR	*p*	Estimation
MDA ← EA	0.027	0.030	0.906	0.365	0.130
AGE ← EA	−0.403	0.194	−2.077	**	−0.289
BMI ← EA	0.177	0.062	2.853	**	0.383
MDA ← EC	−0.009	0.028	−0.312	0.755	−0.045
AGE ← EC	0.396	0.184	2.156	**	0.302
BMI ← EC	−0.024	0.059	−0.413	0.680	−0.056
MDA ← ER	0.027	0.031	0.866	0.386	0.129
AGE ← ER	0.058	0.204	0.285	0.776	0.041
BMI ← ER	0.035	0.065	0.531	0.595	0.074
EC ← → EA	0.332	0.095	3.473	***	0.464
EC ← → ER	0.366	0.096	3.800	***	0.519
EA ← → ER	0.339	0.090	3.748	***	0.510

Note 1: Regression weights (RW); standardized regression weights (SRW); standard error (SE); critical ratio (CR). Note 2: Emotional attention (EA); emotional clarity (EC); emotional Repair (ER); body mass index (BMI); Mediterranean diet adherence (MDA); age (AGE). Note 3: ** *p* ≤ 0.05; *** *p* < 0.001.

## Data Availability

The data used to support the findings of current study are available from the corresponding author upon request.

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
