# Peer review of "Mediterranean Diet Adherence, Body Mass Index and Emotional Intelligence in Primary Education Students—An Explanatory Model as a Function of Weekly Physical Activity"

_children, 2022, doi:10.3390/children9060872_

Round 1

Reviewer 1 Report

The manuscript by Eduardo Melguizo-Ibáñez et al. titled "Mediterranean diet adherence, body mass index and emotional intelligence in primary education students. An explanatory model as a function of weekly physical activity" aimed at establishing the relationship between emotional intelligence, Mediterranean diet adherence, BMI and age. 

This Reviewer considers that the manuscript has many flaws.

  1. Sample size not justified.
  2. Why did not the authors use a validated physical activity questionnaire to assess the level of physical activity? information is not clear about the duration and intensity, physical activity or sport.
  3. The BMI is not a good anthropometric measure for active people. Bioelectrical Impedance should be applied instead.
  4. The explanation given in lines 278-280 is not an explanation for this finding.

Author Response

REVIEWER 1

The manuscript by Eduardo Melguizo-Ibáñez et al. titled "Mediterranean diet adherence, body mass index and emotional intelligence in primary education students. An explanatory model as a function of weekly physical activity" aimed at establishing the relationship between emotional intelligence, Mediterranean diet adherence, BMI and age.

This Reviewer considers that the manuscript has many flaws.

Comment 1

Sample size not justified.

Response 1

Thank you very much for your comment. In this case the sample size is specified in section 2.1 Participants and design.

Comment 2

Why did not the authors use a validated physical activity questionnaire to assess the level of physical activity? information is not clear about the duration and intensity, physical activity or sport.

Response 2

Thank you very much for your comment. In this case, no validated questionnaire has been used since the aim is to measure whether participants practice more than 3 hours of physical activity outside school hours. In this case, the information proposed by the WHO (2020) has been used for the wording of this question. Likewise, the duration and intensity have been omitted since the aim is to study whether the students meet the criteria established above. It should be noted that the following research measures the practice of physical exercise through the question used above:

Melguizo-Ibáñez, E.; Viciana-Garófano, V.; Zurita-Ortega, F.; Ubago-Jiménez, J.L.; González-Valero, G. Physical Activity Level, Mediterranean Diet Adherence, and Emotional Intelligence as a Function of Family Functioning in Elementary School Students. Children 2021, 8, 6. https://doi.org/10.3390/children8010006

Arufe-Giráldez, V.; Zurita-Ortega, F.; Padial-Ruz, R.; Castro-Sánchez, M. Association between Level of Empathy, Attitude towards Physical Education and Victimization in Adolescents: A Multi-Group Structural Equation Analysis. Int. J. Environ. Res. Public Health 2019, 16, 2360. https://doi.org/10.3390/ijerph16132360

The question related to physical activity evaluates the regular practice of physical activity, offering a dichotomous answer ("Yes" and "No"), considering that a subject who practices physical activity three or more hours a week outside of school is physically active, so this criterion has been used, coinciding with international recommendations.

Comment 3

The BMI is not a good anthropometric measure for active people. Bioelectrical Impedance should be applied instead.

Response 3

Thank you very much for your comment. In this case, BMI has been used because the instrument used to perform the bioelectrical impedance could not be moved to the educational center. Also, during data collection (March 2021) the researchers could not access the school due to restrictions imposed to stop the spread of the COVID-19 virus. 

Comment 4

The explanation given in lines 278-280 is not an explanation for this finding.

Response 4

Thank you very much for your comment. This part of the discussion has been redone in response to your suggestions.

Reviewer 2 Report

Dear Authors,

Thank you for your manuscript.

Background. My major concerns are an insufficient theoretical background of your model and incomplete presentation of the study instruments. Looking at Figure 1, I understand that age, adherence to the Mediterranean diet and body mass index (dependent variables) are predicted by the emotional intelligence represented by three subdomains (independent variables). For diet it is ok, but how age and BMI can be predicted by emotional intelligence? Possibly older age can be associated with higher emotional intelligence, while emotional intelligence can lead to a healthier diet and, as a consequence, a more healthy BMI. But BMI being lower or higher doesn't indicate that it is more or less healthy. Including BMI in the model doesn't make sense to me. Literature analysis mostly focuses on separate constructs of the study and lacks explaining theoretical associations between them. I would suggest revising the theoretical background for the model, the model itself and moving it from the Methods to the Introduction section.

Minor comments. Lines 51-53: please revise the sentence "Unfortunately, currently adopted dietary patterns are based on a calorie imbalance, with uncontrolled fat intake leading to an increased incidence of cardiovascular diseases, including obesity". The reader might understand that obesity is one of the CVD. Line 63: please change "intense" to "vigorous".

Methods. Please provide the full names of the scales and questionnaires used in the abstract and Methods section (TMMS-24). Also, as study participants are young, there is a need to justify that the emotional intelligence measure is applicable for the age range 9-13 years.

In section 2.1, please indicate the date when the study was conducted. As in section 2.3. it is stated that "Due to the COVID-19 pandemic, project researchers were unable to personally attend the school" (lines 144-145). But the date is not specified.

The description of the BMI and physical activity should be started in a new paragraph each, as now they seem to be attributed to the sociodemographic factors. Please explain how the question "Do you engage in more than 3 hours of physical activity outside of school?" corresponds to the WHO PA guidelines? Also, please revise the sentence as "hours" is repeated twice. The WHO PA guidelines for kids and adolescents state that:

  1. Children and adolescents should do at least an average of 60 minutes per day of moderate to vigorous-intensity, mostly aerobic, physical activity, across the week. 
  2. Vigorous-intensity aerobic activities, as well as those that strengthen muscle and bone, should be incorporated at least 3 days a week.

According to your question, the classification of the study subjects into less than three hours per week or more seems to be not justified. Was the intensity and type (aerobic, anaerobic) of PA taken into account?

Please provide the percentage of underweight, normal weight and overweight/obese subjects in the sample together with the anthropometric description.

Line 175 - I think you do not use latent variables in this study?

Results. Tables 1-3 are confusing with missing abbreviations explained. The title of Table 1 should indicate the comparison of study measures in the groups defined. Comparative analysis of No and Yes - what does that mean? Levene's statistics are excessive and make the table even more difficult to read. Tables 2 and 3 are unclear. I can not understand the meaning of % MDA and % PA provided. I suggest providing only the percentage calculated from the row and additionally providing the row and the column "Total" so the reader can understand how percentages were calculated. Please provide the explanations of what PA, MDA and ES mean. Statistics illustrating significance should be moved to the footnotes.

The path model should include *, ** or similar marking to indicate the significance of the regression coefficients. Also, for me, the main principle of the multigroup analysis in this paper is unclear. Usually, the model with the path coefficients is provided for the total sample, and then invariance testing is conducted with the aim to test model's difference between groups. As a result, model invariance statistics should be presented.

I will review the discussion and conclusion after these important points will be clarified.

Author Response

Comment 1

Background: My major concerns are an insufficient theoretical background of your model and incomplete presentation of the study instruments. Looking at Figure 1, I understand that age, adherence to the Mediterranean diet and body mass index (dependent variables) are predicted by the emotional intelligence represented by three subdomains (independent variables). For diet it is ok, but how age and BMI can be predicted by emotional intelligence? Possibly older age can be associated with higher emotional intelligence, while emotional intelligence can lead to a healthier diet and, as a consequence, a more healthy BMI. But BMI being lower or higher doesn't indicate that it is more or less healthy. Including BMI in the model doesn't make sense to me. Literature analysis mostly focuses on separate constructs of the study and lacks explaining theoretical associations between them. I would suggest revising the theoretical background for the model, the model itself and moving it from the Methods to the Introduction section.

Response 1

Thank you very much for your suggestion. The authors of the present investigation consider that the justification of the model has not been carried out correctly and therefore we have added to the introduction the justification of the variables according to age and BMI. Also, in response to the suggestion to place the model in the introduction of the article, the authors have decided to move it to the introduction of the article. Regarding the instruments used for data collection, all the questionnaires used are validated and have a high degree of reliability. In this case, for each of the questionnaires, an explanation of the instrument has been carried out, together with Cronbach's alpha, which determines the degree of reliability of the questionnaires.

Comment 2

Minor comments. Lines 51-53: please revise the sentence "Unfortunately, currently adopted dietary patterns are based on a calorie imbalance, with uncontrolled fat intake leading to an increased incidence of cardiovascular diseases, including obesity". The reader might understand that obesity is one of the CVD. Line 63: please change "intense" to "vigorous".

Response 2

Thank you very much for your comment. The requested suggestions have been implemented.

Comment 3

Methods. Please provide the full names of the scales and questionnaires used in the abstract and Methods section (TMMS-24). Also, as study participants are young, there is a need to justify that the emotional intelligence measure is applicable for the age range 9-13 years.

Response 3

Thank you very much for your suggestion. In this case, the full name of the questionnaire and the justification for the use of the questionnaire for data collection in the selected population has been added.

Comment 4

In section 2.1, please indicate the date when the study was conducted. As in section 2.3. it is stated that "Due to the COVID-19 pandemic, project researchers were unable to personally attend the school" (lines 144-145). But the date is not specified.

Response 4

Thank you very much for your comment. In this case, the month together with the year in which the data was collected has been added.

Comment 5

The description of the BMI and physical activity should be started in a new paragraph each, as now they seem to be attributed to the sociodemographic factors. Please explain how the question "Do you engage in more than 3 hours of physical activity outside of school?" corresponds to the WHO PA guidelines? Also, please revise the sentence as "hours" is repeated twice. The WHO PA guidelines for kids and adolescents state that:

  • Children and adolescents should do at least an average of 60 minutes per day of moderate to vigorous-intensity, mostly aerobic, physical activity, across the week.
  • Vigorous-intensity aerobic activities, as well as those that strengthen muscle and bone, should be incorporated at least 3 days a week.

According to your question, the classification of the study subjects into less than three hours per week or more seems to be not justified. Was the intensity and type (aerobic, anaerobic) of PA taken into account?

Response 5

Thank you very much for your comment. In this case, no validated questionnaire has been used since the aim is to measure whether participants practice more than 3 hours of physical activity outside school hours. In this case, the information proposed by the WHO (2020) has been used for the wording of this question. Likewise, the duration and intensity have been omitted since the aim is to study whether the students meet the criteria established above. It should be noted that the following research measures the practice of physical exercise through the question used above:

Melguizo-Ibáñez, E.; Viciana-Garófano, V.; Zurita-Ortega, F.; Ubago-Jiménez, J.L.; González-Valero, G. Physical Activity Level, Mediterranean Diet Adherence, and Emotional Intelligence as a Function of Family Functioning in Elementary School Students. Children 2021, 8, 6. https://doi.org/10.3390/children8010006

Arufe-Giráldez, V.; Zurita-Ortega, F.; Padial-Ruz, R.; Castro-Sánchez, M. Association between Level of Empathy, Attitude towards Physical Education and Victimization in Adolescents: A Multi-Group Structural Equation Analysis. Int. J. Environ. Res. Public Health 2019, 16, 2360. https://doi.org/10.3390/ijerph16132360

The question related to physical activity evaluates the regular practice of physical activity, offering a dichotomous answer ("Yes" and "No"), considering that a subject who practices physical activity three or more hours a week outside of school is physically active, so this criterion has been used, coinciding with international recommendations.

Comment 6

Please provide the percentage of underweight, normal weight and overweight/obese subjects in the sample together with the anthropometric description.

Response 6

Thank you very much for your suggestion. A table showing the descriptive analysis of the BMI variable has been added.

Comment 7

Line 175 - I think you do not use latent variables in this study?

Response 7

Thank you very much for your comment. In this case the proposed model does not present any latent variable, so it has been removed.

Comment 8

Results. Tables 1-3 are confusing with missing abbreviations explained. The title of Table 1 should indicate the comparison of study measures in the groups defined. Comparative analysis of No and Yes - what does that mean? Levene's statistics are excessive and make the table even more difficult to read. Tables 2 and 3 are unclear. I can not understand the meaning of % MDA and % PA provided. I suggest providing only the percentage calculated from the row and additionally providing the row and the column "Total" so the reader can understand how percentages were calculated. Please provide the explanations of what PA, MDA and ES mean. Statistics illustrating significance should be moved to the footnotes.

Response 8

Thank you very much for your suggestion. In this case the authors consider that it helps to improve the understanding of the statistical analysis.

In table 1, the name of the variables has been added to the title and the meaning of the abbreviations has been added in note 2. Likewise, in tables 2 and 3 a % has been deleted and the row has been left, adding the total value and the total sum of the participants who meet the criterion. Finally, with respect to the statistics illustrating significance, the authors consider that leaving them in the tables provides less clutter and less chaos than changing them to the bottom of the page, since they are more easily observed.

Comment 9

The path model should include *, ** or similar marking to indicate the significance of the regression coefficients. Also, for me, the main principle of the multigroup analysis in this paper is unclear. Usually, the model with the path coefficients is provided for the total sample, and then invariance testing is conducted with the aim to test model's difference between groups. As a result, model invariance statistics should be presented.

Response 9

Thank you very much for your suggestion. In this case, the analysis has been carried out for the whole sample, adding a new table and a new figure (Figure 2 and Table 5). In response to your suggestion to add the characters * and ** to the figures of the models, the authors have considered that adding them to the images will overload the images and therefore make them more difficult to understand. In this case, the degree of statistically significant differences is marked in the tables, which act as an element of support and understanding for the proposed equation models.

Reviewer 3 Report

Thank you for the opportunity to review this article. Congratulations to the authors for this study. However, I have also some comments for the authors of this article. 

Introduction. The introduction discusses the benefits of physical activity, Mediterranean diet adherence, emotional intelligence quite clearly and specifically. However, I missed more detail explanation of the hypotheses. Especially for the second hypothesis.

Material and methods. Material and methods were described quite clearly. But I suggest to specify what do you mean by stating that "Schools were selected to participate randomly" (line 104).

Results. The results are good. Just not very clear what it means "sport criterion" (line 191)

Discussion. Discussion mostly focus on comparison of obtained data with the previous findings of others research. What is good, but I think not enough.  I suggest to add also some recommendation based on these study findings. 

Author Response

Thank you for the opportunity to review this article. Congratulations to the authors for this study. However, I have also some comments for the authors of this article.

Comment 1

Introduction. The introduction discusses the benefits of physical activity, Mediterranean diet adherence, emotional intelligence quite clearly and specifically. However, I missed more detail explanation of the hypotheses. Especially for the second hypothesis.

Response 1

Thank you very much for your comment. In this case and according to your suggestion, the introduction has been modified.

Comment 2

Material and methods. Material and methods were described quite clearly. But I suggest to specify what do you mean by stating that "Schools were selected to participate randomly" (line 104).

Response 2

Thank you very much for your comment. In this case the sentence "Schools were selected to participate randomly" means that of all the schools in the city of Granada, those chosen for participation were not chosen by convenience sampling, but by chance.

Comment 3

Results. The results are good. Just not very clear what it means "sport criterion" (line 191).

Response 3

Thank you very much for your comment. In this case the word would be criteria.

Comment 4

Discussion. Discussion mostly focus on comparison of obtained data with the previous findings of others research. What is good, but I think not enough.  I suggest to add also some recommendation based on these study findings.

Response 4

Thank you very much for your question. A final paragraph has been added to the discussion in response to your suggestion.

Round 2

Reviewer 1 Report

There are many flaws in the manuscript that have not been addressed by the authors in their reply.

Author Response

Dear Reviewer, 

Please see the the responses to comments.

Thank you!

Reviewer 2 Report

Dear Authors,

Thank you for taking into account my comments and suggestions. The paper was significantly improved. I find several minor issues to be improved. 

I do not find the anthropometic measurements description and the calculation of the BMI and criteria for classification into under-, normal and overweight groups ir the Methods section. Also, I do not see the statistics of the model on multigroup analysis.

Author Response

Comment 1

Dear Authors,

Thank you for taking into account my comments and suggestions. The paper was significantly improved. I find several minor issues to be improved.

I do not find the anthropometic measurements description and the calculation of the BMI and criteria for classification into under-, normal and overweight groups in the Methods section. Also, I do not see the statistics of the model on multigroup analysis.

Response 1

First of all, the authors would like to thank you for the revisions made to help improve the quality of the article.

In response to your new suggestion, in the material and method section (in green) there is an explanation of how the BMI variable was calculated and the ways of categorizing this variable.

Finally, as you mentioned in your comment, the characteristics of the model had not been specified. In this case, in green, you can see the results obtained for each of the values (lines 292-296).

Thank you!

This manuscript is a resubmission of an earlier submission. The following is a list of the peer review reports and author responses from that submission.

Round 1

Reviewer 1 Report

General comment

The manuscript is within the scope of the journal. However, it needs major revision due to serious methodological drawbacks which were not taken into serious consideration and are not even mentioned as major limitations of the present cross-sectional study. Thus the reliability and validity of the findings  are jeopardized.

Specific comments:

Abstract

  1.  Lines 35-36” “The results show that participants who practice more than three hours of physical activity per week have better emotional intelligence scores than those who do not meet this criterion”. Lines 307-309: However in the discussion section is mentioned that  “In contrast, from the structural equation models it seems that  “better relationships between emotional attention and repair are shown in participants who do not practice more than three hours of physical activity per week”. It seems that the results of the study were drawn only from the first two comparative analyses. However, it also seems that the structural equation models failed to clarify the aforementioned correlations . Please comment
  2. Materials and Methods
  3. Lines 103-104: “A non-experimental (ex post facto), descriptive and cross-sectional design was carried out among students from different public schools of the city of Granada”. How the selection of the public schools has been made? Randomly?  Was it a convenience sample? Please specify the type of sampling procedure. Lines: 318-320: In the limitation, the section is written that “the sample consisted of primary school pupils from a very specific geographical area. This prevents the results from being generalized to wider geographical areas belonging to national or regional regions. Which are the demographic characteristics of the chosen area?
  4.  Who filled the questionnaires? Parents through teachers following the instructions of the researchers? In 2.1 subsection, Lines 107-109 is written:  “A cover letter was sent to the parents informing about the aims and nature of the study”. Lines 142-143: However in the next section is written “Teachers received instructions from researchers to answer the questions satisfactorily”. Please clarify. Please provide more information relevant to the sampling procedure. Which was the response rate? The whole procedure due to the COVID-19 pandemic engages both selection and information bias. How was BMI calculated?  Weight and height were probably self–reported. All this info has to be clearly stated in the text
  5. How do the authors define “physical activity”?  within the school or extracurricular?  please provide more specific information and indicative key questions

Data analysis

  1. Which is the statistical power of the sample size? Was the sample sufficient for the assessment of minimum detectable standardized, two-sided differences at a 5% significance level?

Discussion

  1. It is worth mentioning that in the discussion section no reliable justifications were given in order to support the contradiction observed between the findings of the present study as compared to those concluded in the literature.

For instance : Lines : 268-275 “According to the relationship between BMI and physical activity practice, it is ob- served that students who claim to practice more than three hours of physical activity per week are more overweight and underweight than those who do not meet the aforementioned physical criteria”. The references used by the authors in order to comment on the findings are irrelevant to elementary students (they refer to college students or parents) and do not support or even explain the findings. The obvious effect of over and under-reporting of children is not even mentioned!!

In general, the presentation of the findings of the structural equation models failed to clarify the relationships between emotional intelligence, adherence to the Mediterranean diet, BMI, and age in a  comprehensive manner.

  1. In the limitation sections, the authors should state ALL the limitations of the present study as major weaknesses of the applied methodology.

Author Response

Dear Reviewer, 

Please see the cover letter.

Thank you!

Reviewer 2 Report

Dear authors,

This is an interesting study. In my opinion the manuscript would be significantly improved after a extensive english editing.
In the introduction there are several sentences that need rephrasing (e.g. in L. 53: obesity is not considered a cardiovascular disease but a disease itself and a risk factor).

In the results section the sentences which mention participants as higher or lower are not clear. In general I found the results section difficult to follow and confusing.

Author Response

(The authors gave the same response as above.)

Round 2

Reviewer 1 Report

In the discussion section the main limitations of the study are still missing i.e. low statistical power , self reported height and weight of children.

In the methodology section the statistical power of the sample has to be reported.

Author Response

Comment 1

In the discussion section the main limitations of the study are still missing i.e. low statistical power, self reported height and weight of children.

Response 1

Thank you very much for your comment. In this case, both the constraints and future prospects have been collected from line 326 to 340. The new modifications are marked in yellow.

Comment 2

In the methodology section the statistical power of the sample has to be reported.

Response 2

Thank you very much for your comment. In this case the sampling error is included in lines 109-112, specifically in Participants and design section. In this case, to make it easier to visualize, it is marked in yellow.

Reviewer 2 Report

Dear authors,

I noticed the changes that  you made in your manuscript according to the suggestion of Reviewer 1, which improved the manuscript.
However, I did not notice changes according to my suggestions. The manuscript still needs english editing and the characterisation of children as lower or higher is still present (e.g. are lower than participants who need to improve their diet and do not meet the physical activity criteria (64.5%). In addition, it is also shown that participants who practice more than 3 hours of physical activity per week and have a low- quality diet (85.7%) are higher).

I still believe that the relationship between adherence to the MD, Body Mass Index and emotional intelligence is not presented clearly in the manuscript.

Author Response

Comment 

Dear authors,

I noticed the changes that  you made in your manuscript according to the suggestion of Reviewer 1, which improved the manuscript.
However, I did not notice changes according to my suggestions. The manuscript still needs english editing and the characterisation of children as lower or higher is still present (e.g. are lower than participants who need to improve their diet and do not meet the physical activity criteria (64.5%). In addition, it is also shown that participants who practice more than 3 hours of physical activity per week and have a low- quality diet (85.7%) are higher).

Response 

Thank you very much for your comment. The modification suggested in table 3 has been implemented.

Comment 

I still believe that the relationship between adherence to the MD, Body Mass Index and emotional intelligence is not presented clearly in the manuscript.

Response 

Thank you very much for your suggestion. In this case tables 1, 2 and 3 show a correlational analysis of the variables, however, the existing relationships between the variables can be observed in the equation model proposed. To better understand this relationship, the following hypotheses have been put forward:

  • Participants who reported more than three hours of physical activity per week would reflect better associations between BMI, adherence to the Mediterranean diet and emotional intelligence.
  • Participants who stated that they practice less than three hours of physical activity per week will reflect worse associations between BMI, adherence to the Mediterranean diet and emotional intelligence.

Likewise, the research responds to these two hypotheses, since, if Tables 4 and 5 are observed, it is possible to observe the existing relationships between the variables according to whether the participants practice more than 3 hours of physical activity per week. Furthermore, focusing attention on the concussions, the following is stated:  “it is observed that participants who report more than 3 hours of physical activity per week show better relationships between adherence to the Mediterranean diet and at-tention and emotional clarity, however, participants who do not practice more than 3 hours show better relationships between adherence to the Mediterranean diet and emotional repair. It is also observed that participants who practice less than 3 hours of physical activity per week show better relationships between BMI and emotional re-pair and attention.Finally, focusing attention on the relationship between the variables that make up emotional intelligence, better relationships are observed in participants who practice more than 3 hours of weekly physical activity, however, participants who do not meet this criterion, show better relationships between emotional attention and emotional repair.” The authors consider that the existing relationship between all the variables can be observed in the structural equation models, with Tables 1, 2 and 3 as a support element for the understanding of these relationships. 

Finally, some sentences that were not written in the impersonal have been corrected in terms of wording. In addition, an English native speaker has carried out a revision of the expression.

Round 3

Reviewer 1 Report

The limitations of the study are not mentioned in the appropriate section (covid -19 pandemic is a justification of some of the limitations). The authors had to mention all the limitations

The statistical power was not calculated and was not reported (only sampling error was reported). The very small sample size puts the credibility of the results under question